# Ultra-Fine Ruthenium Oxide Quantum Dots/Reduced Graphene Oxide Composite as Electrodes for High-Performance Supercapacitors

**DOI:** 10.3390/nano12071210

**Published:** 2022-04-04

**Authors:** Jie Zhao, Jianmin Zhang, Hang Yin, Yuling Zhao, Guangxu Xu, Jinshi Yuan, Xiaoyao Mo, Jie Tang, Fengyun Wang

**Affiliations:** 1College of Physics, Qingdao University, No. 308 Ningxia Road, Qingdao 266071, China; zj120513@163.com (J.Z.); hangyinccc@gmail.com (H.Y.); tsuixgx1218@163.com (G.X.); yuanjinshi@foxmail.com (J.Y.); mxyphysics@163.com (X.M.); 2National Engineering Research Center for Intelligent Electrical Vehicle Power System, College of Mechanical and Electrical Engineering, Qingdao University, No. 308 Ningxia Road, Qingdao 266071, China; 3State Key Laboratory of Bio Fibers and Eco Textiles, Qingdao University, No. 308 Ningxia Road, Qingdao 266071, China; zhaoyuling@qdu.edu.cn; 4National Institute for Materials Science, 1-2-1 Sengen, Tsukuba 3050047, Japan

**Keywords:** supercapacitor, ultra-fine RuO_2_ nanoparticles, reduced graphene oxide, nanocomposites, microwave-assisted hydrothermal synthesis

## Abstract

This study synthesized ultra-fine nanometer-scaled ruthenium oxide (RuO_2_) quantum dots (QDs) on reduced graphene oxide (rGO) surface by a facile and rapid microwave-assisted hydrothermal approach. Benefiting from the synergistic effect of RuO_2_ and rGO, RuO_2_/rGO nanocomposite electrodes showed ultra-high capacitive performance. The impact of different RuO_2_ loadings in RuO_2_/rGO nanocomposite on their electrochemical performance was investigated by various characterizations. The composite RG-2 with 38 wt.% RuO_2_ loadings exhibited a specific capacitance of 1120 F g^−1^ at 1 A g^−1^. In addition, it has an excellent capacity retention rate of 84 % from 1A g^−1^ to 10 A g^−1^, and excellent cycling stability of 89% retention after 10,000 cycles, indicating fast ion-involved redox reactions on the nanocomposite surfaces. These results illustrate that RuO_2_/rGO composites prepared by this facile process can be an ideal candidate electrode for high-performance supercapacitors.

## 1. Introduction

In recent decades, many efforts have focused on developing high power and high energy density energy conversion and storage devices for the depletion of fossil fuels and the growing problem of environmental pollution [1,2,3,4]. As an excellent energy storage device, Supercapacitors have drawn great research attention for the past few years [5,6,7]. Compared with batteries, supercapacitors have faster charge and discharge rates, higher power density, and longer cycle life due to their energy storage mechanism of physical adsorption and desorption (lithium-ion batteries are intercalation and deintercalation lithium ions) [8,9,10,11]. However, because the electrode materials are mainly carbon, they still have limitations: low energy density (≤10 Wh·kg^−1^) than batteries (>100 Wh·kg^−1^)and limited specific capacitance is far from being satisfactory to meet the increasing demands in the future applications [12,13,14].The main material of the commercially available capacitors is active carbon, with the standard grade today providing a capacitance of 100 F/g [15]. Therefore, improving the energy density of supercapacitors is a significant research direction.

To increase the energy density, according to the equation (E = 1/2 CV^2^), the energy (E) could be enhanced by increasing the specific capacitance (C) [16,17]. Transition metal oxides, such as RuO_2_, Fe_2_O_3_, MnO_2_, and Co_3_O_4_, have been demonstrated as supercapacitor electrodes, providing a large capacity [18,19,20,21]. Based on retaining carbon-based materials, adding transition metal oxides can increase the specific capacity of the materials through the Faradaic reaction on the electrode surface [22]. However, the electrochemical activity and kinetics of transition metal oxides determine the properties of electrode materials, which often leads to unsatisfactory high-rate capability and reversibility [23,24]. Therefore, by combining carbon materials and transition metal oxides, the design of composite electrode material can make full use of their synergic effects to meet the target of obtaining high electrochemical electrodes for supercapacitors [25,26]. Among the extensively explored transition metal oxides, RuO_2_, with high theoretical specific capacitance and wide potential window, has been investigated as a promising electrode material for supercapacitors [27]. However, due to its low surface area and easy aggregation after reactions, RuO_2_ often suffers from inadequate redox reactions with electrolytes, resulting in relatively low specific capacity and limiting its application as electrodes of supercapacitors. In addition, the poor electrical conductivity of metal oxides also limits its rate capability and cycle stability [28].

To overcome the shortcomings mentioned above and realize the theoretical capacity of RuO_2_, it is of great significance to improve the ion and electron transport kinetics in the electrode and the electrode-electrolyte interface. It can ensure that enough electroactive sites are exposed to the electrolyte to undergo a redox reaction, called the Faradaic redox reaction [29]. To meet these assumptions, some carbon-based materials such as carbon nanofibers, carbon nanotubes, and activated carbon are composited with nanostructured metal oxides to obtain nanostructured electrode materials [30,31,32]. Among the various carbonaceous materials, graphene, having a two-dimensional honeycomb lattice structure, has been considered a promising candidate due to its high surface area, outstanding electrical conductivity, great structural flexibility, and robust mechanical properties [33]. Moreover, the graphene precursor, graphene oxide, has abundant oxygen-containing functional groups on the surface. It serves as site for bonding with metals or metal oxides, making it a precursor for synthesizing graphene and RuO_2_ [34]. After ruthenium oxide and graphene are composited, the graphene in the composite material can provide a three-dimensional conductive network, improve the entire system’s conductivity, and increase the specific surface area [35]. However, these composites usually suffer from RuO_2_ severe nanoparticles (NPs) aggregation, resulting in low capacitance [22].

The microwave-hydrothermal technique has been used extensively to synthesize carbon and RuO_2_ composites. For instance, Yan et al. fabricated RuO_2_ nanoparticles on carbon nanotubes using the microwave-assisted method, showing a capacitance of 494 F/g at a scan speed of 50 mV/s [36]. Hu et al. prepared a composite of RuO_2_ × H_2_O–TiO_2_ using a microwave-assisted hydrothermal method and obtained a capacitance of 992 F/g at 100 mV/s [37]. However, these composites still suffered from significant aggregation. At present, there are very few reports on the synthesis of ruthenium oxide quantum dots (QDs)/graphene composites by the microwave hydrothermal method.

Considering the factors mentioned above, we report a microwave-hydrothermal method to synthesize homogeneous RuO_2_ and graphene nanocomposite (RuO_2_/rGO) and investigate its properties as the electrode material of supercapacitors. The synthesis method has the advantages of fast heating speed, easy control of pressure and temperature, high yield rate, and good homogeneity. Moreover, we also explored the effect of different ruthenium loadings on the electrode capacity. The composite RG-2 with 37.98 wt.% RuO_2_ loadings showed excellent electrochemical characteristics in 1 M H_2_SO_4_ electrolyte, including high specific capacitance (1120 F/g at 1 A/g), outstanding rate capability (937 F/g at 10 A/g), and superior long-term cycling stability (89% after 10,000 cycles at 5 A/g), demonstrating its great potential as electrode material for high-performance aqueous supercapacitors.

## 2. Materials and Methods

### 2.1. Materials

Graphite (≥98%) and ruthenium chloride hydrate were purchased from Shanghai Hushi Laboratorial Equipment Co., Ltd. (Shanghai, China). and Chinese Academy of Sciences Chengdu Organic Chemistry Co., Ltd. (Chengdu, China).

### 2.2. Synthesis of GO, RuO_2_/Reduced Graphene Oxide Nanosheets Nanocomposites

Graphene oxide (GO) was synthesized from graphite by the modified Hummers method. Graphite (500 mg) and NaNO_3_ (500 mg) powders were first mixed in a beaker. H_2_SO_4_ (23 mL, 98%) was then added to the beaker with stirring in an ice-water bath at 3 °C. KMnO_4_ (3 g) was added to the suspension slowly to avoid overheating, and the suspension was stirred at room temperature for 12 h. After adding H_2_O_2_ (3 mL), the mixture was continuously stirred and then centrifuged several times for cleaning. The obtained GO was then dispersed in distilled water by sonication.

1 M ruthenium chloride hydrate in 50 mL of water is thoroughly mixed with the GO solutions (as prepared, GO powder of 50 mg was mixed with 20 mL of water followed by sonication for 1 h) under stirring conditions for 60 min. The graphite oxide powders were exfoliated via ultrasonic vibration to produce GO nanoplatelets. Afterward, the freshly prepared NH_3_·H_2_O solution was used to maintain the pH seven and stirred continuously at room temperature to obtain RuO_2_/rGO precursor solution. Next, the mixture was transferred to a 100 mL microwave digestion tank, sealed tightly, and was placed in a conventional microwave oven. The total reaction time is 10 min and then cooled down in the air. The as-prepared powder was repeatedly filtered and washed several times. Finally, as-prepared RuO_2_-rGO was annealed for 2 h to obtain RG. The composites synthesized with different RuCl_3_·nH_2_O contents (18, 42, 90 mg). The mass fractions of ruthenium chloride were 28%, 38%, and 54%, and were named RG-1, RG-2, and RG-3, respectively.

### 2.3. Instrumentation

X-ray diffraction (XRD) patterns were performed on a Rigaku Smart-lab diffractometer at a scanning rate of 4° per min in the twoθ ranges from 10° to 70° with Cu-Kα radiation (λ = 0.15405 nm) (Rigaku, Tokyo, Japan). Microscope images were recorded digitally with either a scanning electron microscope (SEM, JSM-6480A) (JEOL, Tokyo, Japan) or a transmission electron microscope (TEM, FEI Tecnai F 30) (FEI, Hillsboro, USA). Software (Image pro plus) measured the average size and size distribution of ruthenium oxide quantum dots. In the representative TEM images, 100 particles of different sizes were randomly selected in the region for statistical analysis. N_2_ adsorption/desorption isotherms were obtained on a Micromeritics ASAP Tristar II 3020 apparatus at 77K (ULVAC-PHI, Tokyo, Japan). Pore size distribution was calculated by the density functional theory (DFT) method. X-ray photoelectron spectra (XPS) measurements were carried out on a VG ESCALAB MK II electron energy spectrometer using Mg KR (1253.6 eV) as the X-ray excitation source. Raman spectra were acquired with a confocal laser micro-Raman spectrometer (LABRAMHR, London, UK). The electrochemical measurements were carried out with a Zahner electrochemical workstation. All measurements were performed at room temperature.

### 2.4. Electrochemical Measurements

Electrochemical measurements were implemented with H_2_SO_4_ (1 M) as the electrolyte in a three-electrode configuration. To assemble a working electrode, 4 mg RuO_2_/rGO, 0.5 mg carbon black, and 0.5 mg polytetrafluoroethylene (PTFE) were dispersed in ethanol by ultrasonication for 1 h and coated evenly on nickel foam. The electrode was then dried in a vacuum at 60 °C for 12 h. A platinum wire was used as the counter electrode, and an Ag/AgCl electrode was adopted as the reference electrode. The symmetric supercapacitors were assembled as obtained materials and 1M H_2_SO_4_ as electrolytes. The cyclic voltammetry (CV), galvanostatic charge/discharge, cycle life, and electrochemical impedance spectroscopy (EIS) measurements were carried out using an electrochemical workstation (Zahner, Kronach, Germany).

## 3. Results and Discussion

The RuO_2_/rGO fabrication is schematically illustrated in Figure 1. As we all know, nanoparticles go through two nucleation processes and growth during the growth process [38]. Microwave energy leads to rapid heating of the system and rapid nucleation of RuO_2_ nanoparticles. Since the reaction time is only 10 min, the nanoparticles do not have enough time to grow, so ultra-small RuO_2_ quantum dots are obtained. Due to the ionization of the oxygen-containing functional groups, hydroxyl and carboxyl groups on the surface of GO, the surface of GO is negatively charged and adsorbs positive divalent Ru ions. In a microwave hydrothermal system, trivalent ruthenium ions are oxidized to ruthenium oxide by oxygen-containing functional groups on the surface of graphene oxide. At the same time, graphene oxide loses functional groups and is reduced to graphene (rGO).

### 3.1. Morphology Analysis

The morphologies of RG-1, RG-2, and RG-3, were characterized by transmission electron microscopy (TEM), as shown in Figure 2a–c. The inset is the particle size distribution. All RuO_2_ quantum dots are uniformly distributed on the graphene sheets, and the high-magnification images are shown in Appendix A. With the increase of ruthenium loading, the particle size gradually increased, and the average particle sizes of RG-1, RG-2, RG-3 were 0.8 nm, 1.8 nm, and 2.5 nm, respectively. Figure 2d shows the high-resolution transmission electron microscopy (HRTEM) image of RG-2 (The yellow line marks RuO_2_ QDs), where lattice fringes of the (110) plane could be found with an interplanar spacing of 0.320 nm. Since the diffraction fringes are not obvious, it is presumed that the crystallinity of RuO_2_ is low. It is worth noting that when the ruthenium oxide content is greater than 67%, the quantum dots start to aggregate as indicated in Appendix A.

### 3.2. Structure Analysis

Figure 3a provides the XRD patterns of reduced graphene oxide (rGO), RG-1, RG-2, RG-3, and RuO_2_. It is observed that reduced graphene oxide, which shows the broad diffraction peak at 2θ = 24 °C corresponding to the (002) plane and a weak diffraction peak at 43 °C related to the (101) plane, respectively. This (002) peak confirms that the microwave-assisted synthesis has successfully reduced the graphene oxide. RG-1, RG-2, RG-3, and pure RuO_2_ showed a broad hump and no clear RuO_2_ diffraction peak. There are two possible reasons: on the one hand, according to the Sheller formula: D = Kγ/B·cos θ, where D is the particle size perpendicular to the standard line of plane, K is a constant (it is 0.9), β is the full width at half-maximum of the diffraction peak, θ is the Bragg angle of the peak, and λ is the wavelength of X-ray [39]. The half-peak width is too large due to the too-small particle size. On the other hand, the sample has low crystallinity, consistent with HRTEM.

Raman spectroscopy was investigated for the existing form of the synthesized RuO_2_ QDs and graphene sheets. In Figure 2b, Raman spectra of reduced graphene oxide at 1351 cm^−1^, corresponding to the D-band (disorder and defectivity) and at 1600 cm^−1^, corresponding to the G-band (degree of graphitization) [40]. The ratios (ID/IG) of the peak intensities (ID/IG) of RG-1, RG-2, and RG-3 were 0.89, 0.94, and 0.98, respectively, indicating the disorder of graphene sheets increased with the increase of ruthenium oxide loading. In addition to the D and G bands associated with rGO, the spectra of the composites with different loadings showed three distinct RuO_2_ peaks located at 503, 616, and 682 cm^−1^, respectively. These results confirmed the formation of RuO_2_ crystals [41]. This result is also consistent with TEM.

To confirm the carbon content of RG-1, RG-2, RG-3, thermogravimetric analysis (TGA) is shown in Figure 3c. The first mass loss at 0~230 °C can be attributed to the evaporation of moisture adsorbed on the graphene surface, while the second mass loss at 230~420 °C can be attributed to the depletion of rGO. The mass of hydrous RuO_2_ kept decreasing as the temperature rose till only RuO_2_ left (about 100 °C). The loss of carbon matrix of RuO_2_/rGO was used to estimate the load of RuO_2_.The results show that the mass fractions of RuO_2_ in RG-1, RG-2, and RG-3 are 28%, 38%, and 54%, respectively.

Typical type-IV curves were obtained from the N_2_ adsorption-desorption isotherms (Figure 3d), indicating the presence of mesoporous structures in the composites. The specific surface area calculated by the Brunauer–Emmett–Teller (BET) method is 263, 351, and 349 m^2^/g for RG-1, RG-2, and RG-3, respectively. It can be concluded that the content of rGO significantly influences the specific BET surface area. In contrast, the composite structure forms more pore structures due to the intercalation of ruthenium oxide quantum dots between graphene sheets. The pore size distribution is shown in Figure 3e. The pore size distribution curve of RG-1, RG-2 and RG-3 is obtained from adsorption branch by the Barret–Joyner–Helena (BJH) analyses. The RG-1, RG-2, RG-3 had a portion of mesopores with a pore size distribution between 1.7 and 40 nm and average pore size of 3.8, 5.5, and 4.3 nm. This pore size distribution provides an effective transmission route for the internal voids during charging and discharging and can improve the electrochemical performance of electrode materials [42].

The XPS analysis was performed to confirm further the surface chemistry of the RuO_2_/rGO nanocomposite (Figure 4a), which revealed the presence of C, O, Ru, and N were investigated. The peaks of Ru3d, Ru 3p_1/2,_ and 3p _3/2_ at 281.3eV, 463 eV, and 485 eV from RuO_2_ were observed. The peak of C 1s is attributed mainly to graphene. The Ru 3d spectrum, as shown in Figure 4b, having two peaks at 286.3 and 282.1 eV attributed to Ru 3d_3/2_ and Ru 3d_5/2_ spin-orbit peaks of RuO_2_, respectively, confirmed the formation of RuO_2_ nanoparticles on the surface of rGO sheets. Figure 4c displays the spectrum of O 1s of RG-2. The O 1s spectrum contains four components at 534.5, 532.8, 531.2, and 529.7 eV, assigned to the C-O, C=O, Ru-OH, and Ru-O components, respectively. It has been reported that a few residual functional groups can be used to obtain stable and highly dispersed nanoparticles [43].

### 3.3. Electrochemical Performance of Supercapacitor

To investigate the electrochemical performance of the RuO_2_/rGO composite in a high-performance supercapacitor, we explored its supercapacitive electrochemical properties in 1M H_2_SO_4_ aqueous electrolyte using the three-electrode system with a platinum wire counter electrode and an Ag/AgCl reference electrode. Figure 5a shows the cyclic voltammetry (CV) curves of the RG-1, RG-2, RG-3 electrodes in the potential range of 0 to 1.0 V with a scan speed of 10 mV/s. All CV curves of the RuO_2_/rGO composite exhibit redox peaks. These redox peaks are attributed to the oxidization and reduction of RuO_2_. Compared with RG-3 and RG-1, RG-2 has a larger integration area and exhibits higher specific capacity. It is speculated that because RG-2 has a suitable ruthenium oxide loading and uniform size distribution of quantum dots, a larger specific surface area can expose more active sites. Therefore, RG-2 has a better specific capacity.

The peak current density on the CV curve will rapidly increase with the scan rate shown in Figure 5b, and the peak potential changes little. All curves maintain their shape at high scan speed, which is attributed to the improved mass transport and electron conduction that should also help improve the rate performance. Due to the ultra-small particles and stable three-dimensional structure, RuO_2_/rGO electrodes can provide shorter electron transport paths, faster ion transport rates, and faster surface Faradaic redox reactions. This accelerates the electrode reaction and leads to an increase in the CV current density. The electrical conductivity of the composites will be illustrated in the following EIS results.

Figure 5c shows the galvanostatic charge/discharge (GCD) curves of RG-2 at different current densities. The current density ranges from 1 A/g to 10 A/g. The deviation from a straight line is attributed to the redox reaction of the charge storage reactions. Moreover, the bent part of the charge-discharge curve corresponds to the redox peak of the CV curve, indicating that RuO_2_ provides reversible Faraday capacitance. As the current density increases, the charge-discharge time gradually decreases, which is probably related to the rate of redox reductions cannot keep abreast of rapid changes in potential and inadequate accommodation of electrolyte ions in the electrode.

To evaluate the effect of graphene sheets on the electrochemical properties of RuO_2_/rGO, the supercapacitor was tested at various charging rates. Figure 5e shows the rate performance of nanocomposites using the GCD measurement under various current densities. As a comparative experiment, the CV curve and charge-discharge curve of rGO are shown in Appendix A. Due to the van der Waals forces, rGO possesses lower capacitance retention with increasing current density. For the RuO_2_/rGO composite, all the specific capacitances of RG-1, RG-2, RG-3 were high. It is worth mentioning that the RG-2 composite exhibits a specific capacitance as high as 1120 F/g at 1 A/g and maintains 937 F/g at a scan rate of 10 A/g, showing the best rate performance. It is speculated that RG-2 exhibits the most excellent performance due to its largest specific surface area, suitable pore size distribution, and uniformly distributed RuO_2_ QDs of appropriate size, which can also undergo rapid redox reactions on the surface under high current.

Electrochemical impedance spectroscopy (EIS) measurements were also carried out at a frequency range from 0.01 to 100 kHz to reflect the electrical conductivity of RG-1, RG-2, and RG-3. It is well established that the electrochemical reaction impedance of the electrode material changes in the semicircle diameter (high-frequency region) of the EIS curve, and the low-frequency region (straight-line part) represents the diffusion rate of ions in the electrolyte [44]. The obtained Nyquist plots are presented in Figure 5d, the semicircle diameters of all these curves are small, and the slopes of the straight lines are all greater than 1, indicating light electrochemical polarization and good capacitive characteristics. Notably, we can find that the RG-2 composite shows a smaller semicircle and a larger tilt angle than RG-1 and RG-3, suggesting that the RG-2 composite has slower diffusion to the electrolyte and faster ion transport than that of RG-1 and RG-3. The EIS results for RG-2 are attributed to the higher specific surface area of RG-1 and RG-3, which endows the RuO_2_ QDs with facilitated charge transfer properties.

Long-term cycling stability test to evaluate the practical application of RG-2 and pure RuO_2_. The charge-discharge test between 0 and −1.0 V was repeated for 10,000 cycles at 5 A/g. As shown in Figure 5f, RG-2 retains 89% of the initial capacitance after repeated 10,000 charge-discharge cycles, which is better than pure RuO_2_ (70%). The capacitance retention rate of the composite material at 10,000 cycles is almost the same as that of the capacitors assembled with commercial activated carbon [12]. The ultra-small RuO_2_ quantum dots and graphene composite structure can provide excellent cycling stability and a stable three-dimensional conductive network, ensuring that RuO_2_/rGO material is a qualified electrode candidate for energy storage applications.

## 4. Conclusions

In summary, we have developed a novel and facile strategy to fabricate RuO_2_/rGO nanocomposite via a one-step microwave hydrothermal method. RuO_2_ QDs of 0.8–2.5 nm was uniformly distributed on rGO to form porous nanocomposites. The nanocomposite has a high specific capacity, rate performance, and excellent cycling stability as a supercapacitor electrode at high current densities. The superior electrochemical performance of the RuO_2_/rGO composite is attributed to the following beneficial factors: (1) Synergistic effect. The RuO_2_ QDs ensured the separation of the rGO sheets, and the rGO sheets also prevented the agglomeration of the RuO_2_ QDs, thereby inhibiting the aggregation of the composites during electrochemical discharge/charge and improving the cycling stability; (2) In the as-prepared RuO_2_/rGO composites, due to the sufficiently small size of its RuO2 QDs, it can reduce the path length of ion and electron transport and provide a large specific surface area, which is conducive to the rapid growth of Redox reaction, which helps to improve its rate capability; (3) The intense contact between RuO_2_ QDs and rGO sheets can ensure the structural stability of RuO_2_/rGO composites, help to inhibit the exfoliation of RuO_2_ QDs and improve the electronic conductivity; (4) The porous network structure of RuO_2_/rGO composite can provide abundant active sites in contact with the electrolyte, and can also promote the diffusion of the electrolyte in the electrode material, which is beneficial to achieve higher reversible capacity. Therefore, this study provides an efficient and green synthesis method is proposed to realize the effective composite scheme of ultrafine metal oxides and graphene, which has good development prospects in practical applications.

## Figures and Tables

**Figure 1 nanomaterials-12-01210-f001:**
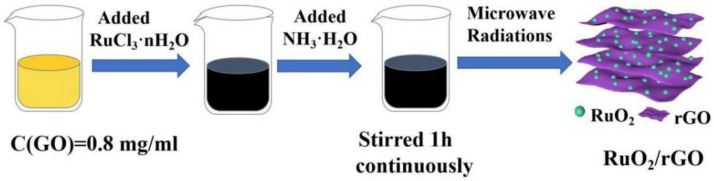
Schematic diagram of RuO_2_ nanoparticles anchored on reduced graphene oxide.

**Figure 2 nanomaterials-12-01210-f002:**
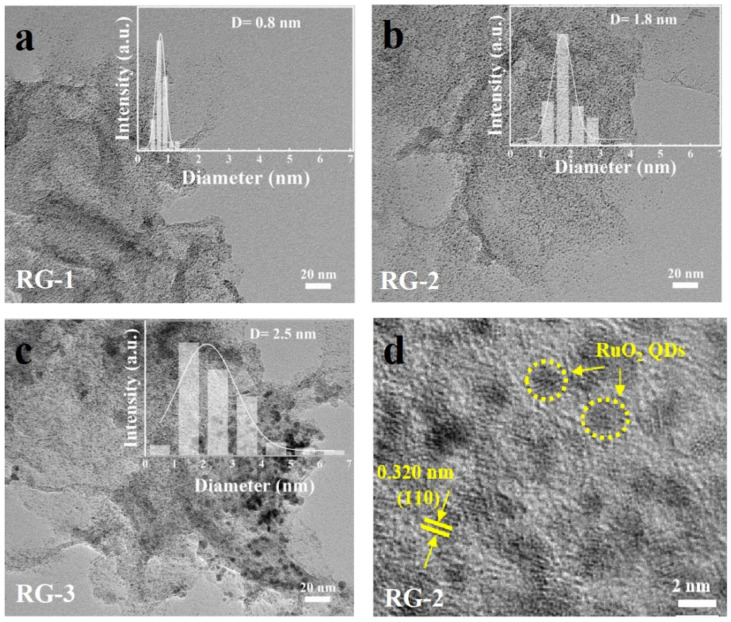
(**a**–**c**), TEM images of RG-1, RG-2, and RG-3, the inset shows the particle size distribution of RuO_2_ QDs. (**d**) HRTEM image of RG-2.

**Figure 3 nanomaterials-12-01210-f003:**
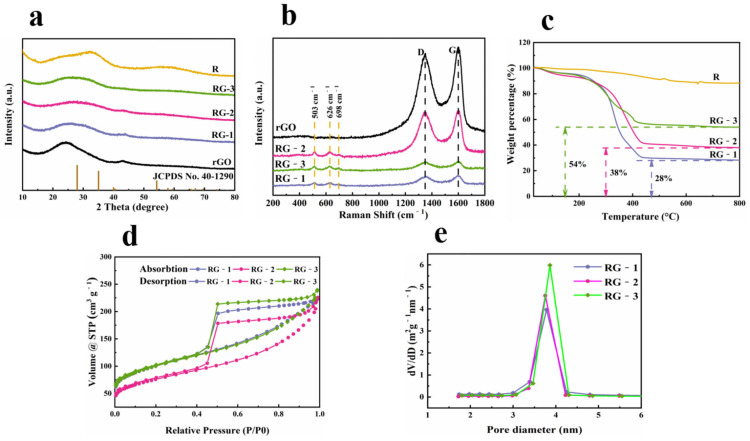
(**a**) XRD patterns of reduced graphene oxide, RG-1, RG-2, RG-3, and RuO_2;_ (**b**) Raman spectra of rGO, RG-1, RG-2, and RG-3; (**c**) TGA curves of RuO_2_, RG-1, RG-2, and RG-3; (**d**) N_2_ adsorption-desorption isotherms and (**e**) Pore size distribution of RG-1, RG-2, and RG-3.

**Figure 4 nanomaterials-12-01210-f004:**
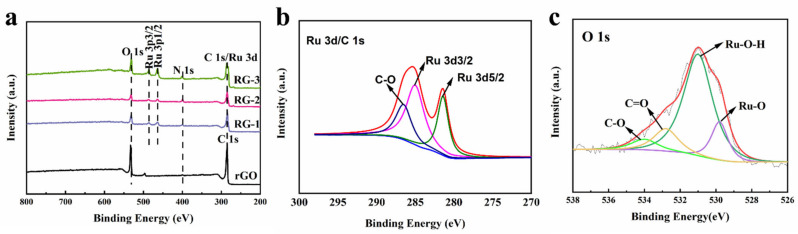
(**a**) XPS spectrum of RG-1, RG-2, RG-3, and rGO; (**b**) Ru 3p/C 1s, (**c**) O1s spectrum of RG-2.

**Figure 5 nanomaterials-12-01210-f005:**
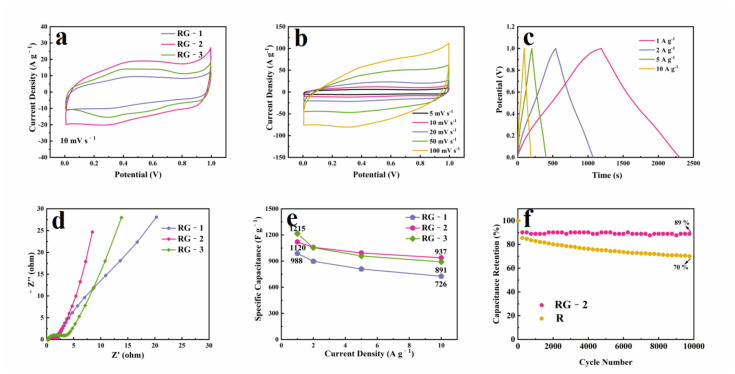
(**a**) CV curves measured at ten mV s^−1^ of RG-1, RG-2, and RG-3; (**b**) CV curves at different scan rates of the RG-2; (**c**) GCD curves at different current densities of the RG-2; (**d**) EIS spectra of RG-1, RG-2, and RG-3; (**e**) the specific capacitance and capacitance retention at different current densities of RG-1, RG-2, and RG-3; (**f**) cycling test of RG-2 and pure RuO_2_.

## Data Availability

Data is contained within the article or Appendix A.

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
