# Peer review of "Ultra-Fine Ruthenium Oxide Quantum Dots/Reduced Graphene Oxide Composite as Electrodes for High-Performance Supercapacitors"

_nanomaterials, 2022, doi:10.3390/nano12071210_

Round 1

Reviewer 1 Report

Report on (nanomaterials-1635398)

"Ultra-Fine Ruthenium Oxide Quantum Dots/Reduced Graphene Oxide

Composite as Electrodes for High-Performance Supercapacitors"

by Jie Zhao, Jianmin Zhang Hang Yin, Yuling Zhao, Guangxu Xu, Jinshi

Yuan, Xiaoyao Mo, Jie TANG, Fengyun Wang

In this paper, the authors address the problem of the synthesis and characterization of ruthenium oxide quantum dots on reduced graphene oxide surfaces. A rapid and microwave-assisted hydrothermal method has been first developed which allows for producing RuO$_2$/rGO nanocomposites with various RuO$_2$ loading. The excellent cycling stability, high specific capacity and rate performance of the produced nanocomposites have been then characterized by means of several experimental techniques.

It is an interesting manuscript that deserves to be published in Nanomaterials, once the following points have been addressed:

Point 1: the Authors state that they control the mass fraction of RuO$_2$ and that the QDs are uniformly distributed on rGO. Can the Authors control this distribution such that they can make it non-uniform? Can they also control the shape of the QDS. Does the QD shape has any influence on the electrode performances?

Point 2: do the Authors observe any stress and deformation in the nanocomposites due to QDs? If yes, do they observe any variation of their capacitive performance due to stress?

Author Response

Dear reviewer,

Thank you for your kind letter regarding the manuscript “Ultra-Fine Ruthenium Oxide Quantum Dots/Reduced Graphene Oxide Composite as Electrodes for High-Performance Supercapacitors.” We sincerely appreciate the reviewers’ approval and constructive comments on our manuscript. These comments have helped us improve the manuscript’s quality both in-depth and in detail. All the improvements on the paper have been integrated into the revised version of the manuscript.  

We hope this revised manuscript has addressed your concerns and look forward to hearing from you.

The specific comments by both reviewers have been answered point-by-point. In this reply, the reviewers’ comments are in black fonts, and the answers are in blue fonts.

Reviewer 2 Report

The manuscript presents a graphene oxide/RuO2 composite material envisioned as an electrode for high-performance aqueous supercapacitors. Overall, I found the presentation interesting and compelling, however there are a few areas that must be addressed or can be improved.

 In the introduction, you should list a capacitance range for the best currently commercially available capacitors to give the reader a point of reference.

 At the end of the synthesis description are listed three mass fractions of ruthenium chloride but the preceding description does not describe three synthesis variations. Please describe the different synthesis steps used to obtain these different mass fractions.

 The somewhat random values reported for the mass fractions (ie. 16%, 38% and 81%, rather than 15%, 40%, and 80%) suggest that the resulting mass fractions were measured rather than assumed based on synthesis procedures. Please describe how these measurements were carried out.

 Particle size distributions are show inset in figure 2, however, it is not described how these distributions were obtained. Based on the number of particles present in the images this was most likely done using a software program or custom code implementation. Please provide a detailed description, as well as how many particles were included in the analysis to provide a sense of how statistically representative these distributions are.

 The bin width and x-axis scales on these particle size distributions change from one to the next. It would be helpful to the reader if these were plotted with the same scale and bin widths and perhaps even plotted together for a more direct comparison. At first glance it looks like the distribution in figure 2(c) is showing a smaller particle size than the other two but in fact these are the largest particles.

 The manuscript does not describe how the pore size distributions shown in figure 3(e) were determined.

 Long term cycling stability over 10000 cycles is shown in figure 5(f) at 89% of the initial capacitance. Can the authors also present “typical” performance metrics for commercially available capacitors to give a sense of how this performance compares.

 Some sentences are awkwardly worded (for example in the conclusion “Therefore, this study provides.” is not a complete sentence) which detract from the overall quality of the presentation. While I found I could understand the intent of the authors, it could be improved.

Author Response

(The authors gave the same response as above.)
